# Ape duos and trios: spontaneous cooperation with free partner choice in chimpanzees

Malini Suchak[1,2,3], Timothy M. Eppley[1], Matthew W. Campbell[1] and Frans B.M. de Waal[1,2]

[1] Living Links, Yerkes National Primate Research Center, Lawrenceville, GA, USA
[2] Department of Psychology, Emory University, Atlanta, GA, USA
[3] Department of Animal Behavior, Ecology & Conservation, Canisius College, Buffalo, NY, USA

## ABSTRACT

The purpose of the present study was to push the boundaries of cooperation among captive chimpanzees (*Pan troglodytes*). There has been doubt about the level of co-operation that chimpanzees are able to spontaneously achieve or understand. Would they, without any pre-training or restrictions in partner choice, be able to develop successful joint action? And would they be able to extend cooperation to more than two partners, as they do in nature? Chimpanzees were given a chance to cooperate with multiple partners of their own choosing. All members of the group ($N = 11$) had simultaneous access to an apparatus that required two (dyadic condition) or three (triadic condition) individuals to pull in a tray baited with food. Without any training, the chimpanzees spontaneously solved the task a total of 3,565 times in both dyadic and triadic combinations. Their success rate and efficiency increased over time, whereas the amount of pulling in the absence of a partner decreased, demonstrating that they had learned the task contingencies. They preferentially approached the apparatus when kin or nonkin of similar rank were present, showing a preference for socially tolerant partners. The forced partner combinations typical of cooperation experiments cannot reveal these abilities, which demonstrate that in the midst of a complex social environment, chimpanzees spontaneously initiate and maintain a high level of cooperative behavior.

## INTRODUCTION

Cooperation, joint action by two or more individuals to achieve a goal, is often regarded as less puzzling than altruistic behavior, in which one individual benefits another at a cost to himself or herself. While this may be true in an evolutionary sense, on a proximate level, cooperation often consists of a series of potentially complex decisions involving a choice of partners. When multiple potential partners are available, an individual faces many questions: Whom to cooperate with? Has that individual been a good partner in the past? How much to invest in this partner and what to expect in return? Will cooperation yield more benefits than solitary action?

Corresponding author
Malini Suchak,
suchakm@canisius.edu

These questions highlight the complexity of cooperative behavior, and yet for such a seemingly complex phenomenon it is surprisingly ubiquitous across a wide variety of taxa (*Dugatkin, 1997*; *Gadakar, 2006*). This paradox has inspired research examining the emergence and maintenance of cooperative behavior at both the ultimate and proximate levels of explanation. In particular, cooperation among nonhuman primates has attracted considerable research because of the evolutionary implications that such research has for human behavior and the ubiquity of cooperation among wild primates, including coalition formation, food sharing, group hunting, and territorial defense (*de Waal & Suchak, 2010*; *Mitani, 2006*; *Muller & Mitani, 2005*). Nevertheless, we know little about the proximate mechanisms of primate cooperation. Do primates coordinate their behavior in space and time? Do they keep track of favors given and received? Do they understand whether and how their partners contribute to successful outcomes? Or do they just simultaneously pursue the same goal? Conceivably, the appearance of cooperation could be created by parties focused entirely on their own individual gain (*Stanford, 1998*). Given the ambiguity of the field data, experimental studies of cooperation have focused on elucidating the underlying cognitive and social mechanisms.

Most experimental work on cooperation has examined cooperation within pre-arranged pairs. Coordinated lever-pressing studies required two monkeys to simultaneously press levers or pull handles to receive food (*Chalmeau, 1994*; *Chalmeau, Visalberghi & Gallo, 1997*; *Visalberghi, Quarantotti & Tranchida, 2000*). These studies mostly demonstrated conditioned responses without an understanding of the contingencies: both individuals continually and rapidly pressed the lever and occasionally succeeded by coincidence (*Visalberghi, Quarantotti & Tranchida, 2000*). In contrast, when two individuals were required to pull in a weighted tray too heavy for one individual, several primate species demonstrated an ability to coordinate pulling rather than instrumental conditioning (*Crawford, 1937*; *Cronin, Kurlan & Snowdon, 2005*; *de Waal & Berger, 2000*; *Mendres & de Waal, 2000*). Subjects demonstrated an understanding of the need for a partner: a juvenile chimpanzee (*Pan troglodytes*) would recruit a partner through gestures (*Crawford, 1937*), cotton-top tamarins (*Saguinus oedipus*) pulled more when a partner was present than when a partner was absent (*Cronin, Kurlan & Snowdon, 2005*), and brown capuchin monkeys (*Cebus apella*) were unsuccessful if they could not see each other, suggesting visual coordination (*Mendres & de Waal, 2000*). The high degree of success in these tasks, as compared to lever-pressing studies has been attributed to the intuitive nature of pulling in a tray baited with food (*Mendres & de Waal, 2000*). In weighted tray tasks, individuals can clearly see the results of their actions and the role of their partner.

Similarly, primates are quite successful at cooperative string-pulling tasks, which require two individuals to simultaneously pull a loose string to bring in a tray of food (Hirata 2003, as cited in *Hirata & Fuwa, 2007*). These studies differ from the weighted tray and lever-pressing studies in that responding before a partner causes the string to release, rendering both subjects unable to solve the task. The critical test is a delay test in which one individual arrives at the apparatus before the other. Understanding of the cooperative nature of the task requires this individual to wait for the second individual before pulling.

This is precisely what has been found in several mammal species (chimpanzees: *Hirata & Fuwa, 2007*; *Melis, Hare & Tomasello, 2006a*; hyenas (*Crocuta crocuta*): *Drea & Carter, 2009*; elephants (*Elephas maximus*): *Plotnik et al., 2011*). Not all species pass this critical test: two bird species able to pull simultaneously, failed to wait for their partner in the delay task (rooks (*Corvus frugilegus*): *Seed, Clayton & Emery, 2008*; parrots (*Psittacus erithacus*): *Péron et al., 2011*).

A limitation of most experimental studies is the elimination of partner choice. This holds for virtually all primate studies (e.g., *Crawford, 1937*; *Hirata & Fuwa, 2007*) but also for experiments on non-primates, such as elephants or birds (*Seed, Clayton & Emery, 2008*; *Plotnik et al., 2011*; *Péron et al., 2011*). An exception is the work by *Melis, Hare & Tomasello (2006b)* and *Melis, Hare & Tomasello (2008)*, which allowed a choice between two potential partners and demonstrated that chimpanzees differentiate between them based on social tolerance and past success. Yet, the partner choice presented in these experiments was still greatly limited compared to the options within an open group setting.

Studies that have allowed open partner choice have generally not found high degrees of cooperation (*Burton, 1977*; *Chalmeau, 1994*; *Chalmeau & Gallo, 1996*; *Chalmeau, Visalberghi & Gallo, 1997*; *Fady, 1972*; *Petit, Desportes & Thierry, 1992*). In fact, the only species to succeed on such a task were Tonkean macaques (*Macaca tonkeana*; *Petit, Desportes & Thierry, 1992*). Although chimpanzees and capuchins succeed at dyadic pulling tasks (*Melis, Hare & Tomasello, 2006a*; *Mendres & de Waal, 2000*), and are known for cooperative behavior in nature (reviewed by *de Waal & Suchak, 2010*), both species failed to establish cooperation in studies offering free partner choice (*Chalmeau, 1994*; *Chalmeau & Gallo, 1996*; *Chalmeau, Visalberghi & Gallo, 1997*). Two possibilities are raised by these results: first, tolerance may be so constrained in the group setting that it prevents cooperation without experimenter interference. Although this idea is supported by the fact that the highly tolerant Tonkean macaques are the only species that succeeded at the task (*Petit, Desportes & Thierry, 1992*), if true partner choice is available, individuals should simply be able to avoid intolerant partners. A second possibility is that the design of the tasks, with one small, highly monopolizable food source as the reward contributed to a contest competition in which there was not an overall net benefit for all participants (e.g., *Chalmeau, 1994*; *Chalmeau, Visalberghi & Gallo, 1997*).

In the current study we allow all individuals access to the apparatus while choosing their own partners. To find out which partners chimpanzees prefer to cooperate with, we carried out experiments in a large outdoor enclosure with the entire group present. All chimpanzees could potentially participate in the cooperative task. We explored several determinants of partner choice. If social closeness were the primary constraint on partner choice (*Melis, Hare & Tomasello, 2006a*), then chimpanzees should work mostly with kin or nonkin affiliates. Closeness in dominance rank may also play a role in cooperation, since closely ranked individuals have similar abilities and needs and may be in the best position to benefit each other (*de Waal & Luttrell, 1986*; *Muller & Mitani, 2005*). Cercopithecine monkeys and chimpanzees who are close in rank tend to interact more than those at greater rank distances (*Silk, 1982*; *de Waal, 1991*; *Muller & Mitani, 2005*). Rank distance

 

**Table 1** Subjects are listed in rank order, with age and any maternal kinship relations provided. Rank was determined using pant grunts, a unidirectional submissive signal. Alpha male and female are denoted with an $\alpha$.

| Chimpanzee | Sex | Age | Rank | Kin |
|---|---|---|---|---|
| SK$\alpha$ | M | 24 | 1 | DN |
| GG$\alpha$ | F | 31 | 2 | BO, RI, KT, TA |
| RN | F | 24 | 3 | |
| BO | F | 47 | 4 | GG, RI, KT, TA |
| MA | F | 47 | 5 | MS |
| KT | F | 22 | 6 | BO, GG, RI, TA |
| AJ | F | 31 | 7 | |
| RI | F | 24 | 8 | BO, GG, KT, TA |
| DN | F | 21 | 9 | SK |
| TA | F | 16 | 10 | BO, GG, KT, RI |
| MS | F | 18 | 11 | MA |

may also play an important role when there is competition for resources (e.g., *de Waal, 1986*), as individuals who are close in rank tend to be more tolerant of each other's presence near a clumped resource. If the presence of the group in the current cooperation task engenders competitive tendencies, then rank distance is expected to affect partner choice. Finally, an alternative hypothesis unrelated to social relationships is that chimpanzees will preferentially choose to work with those with whom they have enjoyed previous successes (*Melis, Hare & Tomasello, 2006b*).

In addition to allowing partner choice, the current study further deviates from the previous work by testing both dyadic and triadic cooperation. In nature, chimpanzee cooperation often requires working with more than one other individual, including coalitions, group hunting and territorial defense (*Muller & Mitani, 2005*). In Kibale National Park in Uganda, for example, male chimpanzees hunt in groups and an increasing number of hunters leads to increased success even though this effect plateaus at six individuals (*Mitani & Watts, 1999*).

Finally, since wild female chimpanzees do not typically participate in cooperative efforts due to the nature of their fission–fusion society, relatively little is known about the dynamics of female–female cooperation. The current study allows us to explore how female chimpanzees choose and switch partners when freed from their natural ecological constraints. Since females of the closely related bonobo are highly cooperative (*Furuichi, 2011*; *Hare et al., 2007*), examining cooperation in female chimpanzees can help shed light on the evolution of cooperative tendencies.

## METHODS

### Subjects and housing

Subjects were 11 chimpanzees (1 male, 10 females, Table 1) kept in a large outdoor enclosure at the Field Station of the Yerkes National Primate Research Center (YNPRC). The group's 711 m$^2$ enclosure contained a large climbing structure and several enrichment

items (barrels, tires, etc.) and was adjacent to indoor sleeping quarters. Testing occurred in the outdoor enclosure with the entire group present and did not require separating individuals from the group. During testing, subjects had access to the indoor sleeping quarters. Chimpanzees were fed two daily meals consisting of fruits, vegetables and grains at approximately 8h30 and 15h00 and had access to water and primate chow *ad libitum*. All food used in this study was supplemental to the chimpanzees' daily intake and at no time was food or water restricted. The chimpanzees were not distressed and were free to stop participating at any time. All procedures were approved by Emory University's Institutional Animal Care and Use Committee (IACUC), protocol #YER-2000180-53114GA. The Yerkes National Primate Research Center is furthermore fully accredited by the Association for Assessment and Accreditation of Laboratory Animal Care (AAALAC).

## Apparatus

The apparatus required that one chimpanzee (in the dyadic condition) or two chimpanzees (in the triadic condition) remove a barrier in order for another chimpanzee to simultaneously pull in a tray baited with food (Fig. 1). The second barrier and the corresponding pull bar were only present in triadic tasks, providing a clear cue to the chimpanzees that a third individual was needed. Once the tray was pulled in all the way (approximately 30 cm) food rewards dropped into a funnel, which delivered them directly to each of the chimpanzees that solved the task. The rods to pull down the barriers and pull in the tray were sufficiently far apart (~1.6 m) so that one individual could not participate in both roles at the same time. If a barrier rod was pulled and released before the tray was pulled in, the barrier rose back into place. Hence, a lone individual could not remove the barrier and pull in the tray; simultaneous action by multiple chimpanzees was required. Food rewards (one grape, two raisins, a small slice of sweet potato or a small slice of banana) varied randomly from trial to trial to maintain the chimpanzees' interest; for each trial all chimpanzees received the same reward. These were not rewards that the chimpanzees received outside the experimental context, thus further increasing their motivation to participate in the task.

## Procedures

### Test sessions

A trial began when the tray was baited with food. Following successful trials, the experimenter waited for the chimpanzees to release the rods and then reset the tray back in the starting position and re-baited the tray. The tray was re-baited as long as the chimpanzees present were not pulling; they did not need to leave and re-approach between trials. If the chimpanzees did not solve the task within 5 min, the trial was considered a failure, the food removed, and a one-minute time out commenced prior to re-baiting. Each session lasted 1 h and consisted of as many trials as could be accomplished in that time period. Only one session was run per day and only 2–3 sessions were run per week to maintain a high degree of motivation. Sessions were run from May 2011 through February 2012.

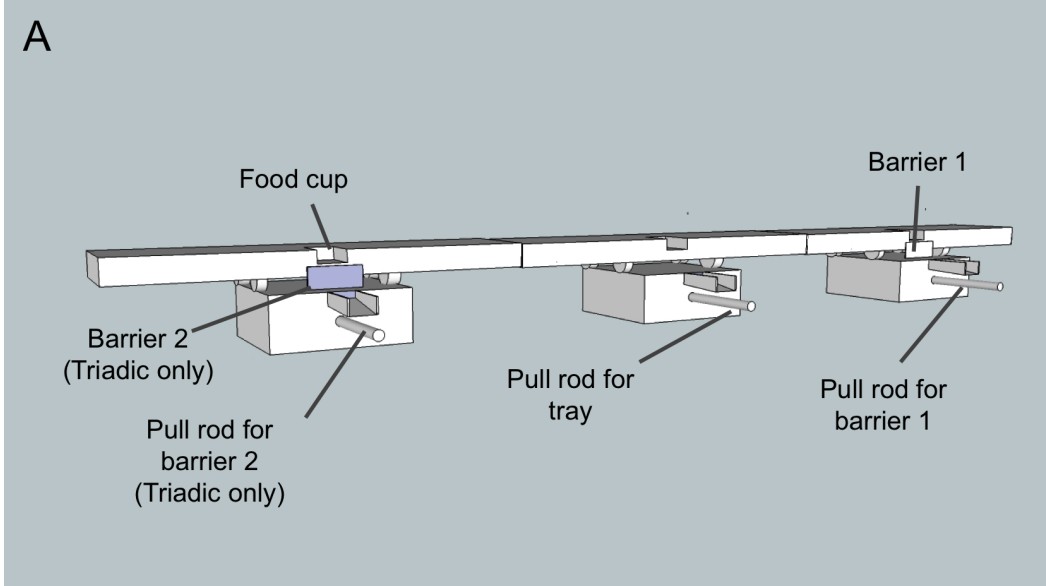

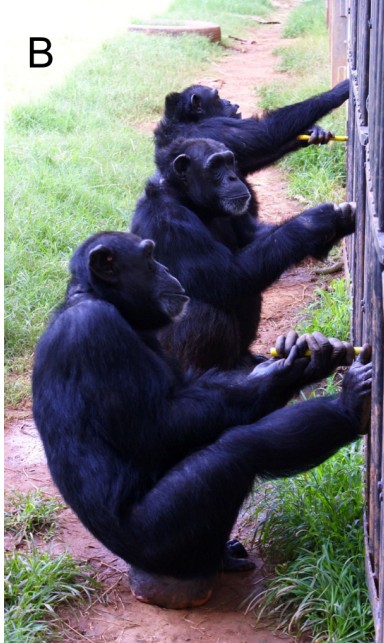

**Figure 1** **The test apparatus and set-up.** (A) Each barrier was connected to a steel rod that extended 20 cm into the chimpanzee enclosure. Pulling on the rod caused the barrier to drop down via a spring/pulley mechanism. Once the barrier (or barriers in the triadic condition) was pulled down a second individual used a similar rod (also extending 20 cm into the enclosure) to pull in the whole tray. The only part of the test apparatus that was inside the enclosure were the pull bars, the rest was outside. Note that barrier 2 (on the left) was only present during the triadic sessions. The apparatus was set up exactly the same for dyadic sessions but barrier 2 and the corresponding pull bar were missing. (B) Three chimpanzees participating in the triadic cooperation task.

## Test phases

### Phase 1a: Dyadic cooperation acquisition

There were 28 dyadic cooperation tests which required two chimpanzees to work together to pull in the tray (one to hold down the barrier and a second to pull in the tray). As there was no training, and none of the chimpanzees had participated in cooperative pulling tasks before, we waited for a significant majority (9 out of 11 chimpanzees, binomial test $p = 0.02$) of the chimpanzees to reach at least 20 successes before moving onto the next phase.

### Phase 1b: Triadic cooperation acquisition

Following Phase 1a a second barrier was added so that three chimpanzees were required to solve the task. There were 28 triadic cooperation tests to allow for direct comparison to the acquisition of dyadic cooperation.

### Phase 2: Alternating, proficiency tests

In order to see if proficiency and experience with triadic cooperation influenced partner choice, we began alternating dyadic and triadic sessions. There were 38 alternating sessions, or 19 of each dyadic and triadic.

## Behavioral coding

Each trial was videotaped from two angles (an overview from above, taken from an observation tower, and a front view) using HD digital video cameras. Additionally, one experimenter had a digital voice recorder to record a narrative of any social interactions that occurred during testing. Success or failure of each trial, which chimpanzees solved the task, and which chimpanzees received rewards were recorded in-person and later confirmed from video. Latency to succeed and the number of pulls before success were also recorded from video. Pulling included any movement of the barrier as well as any bodily pulling motion. A second rater coded a subset of the videos and the two ratings were highly correlated for both latency ($r = 0.99, p < 0.001$) and pulling ($r = 0.85, p < 0.001$). Agreement was also excellent for the identities of the chimpanzees participating in the task (Cohen's Kappa $= 0.89$) and success at the task (Kappa $= 1$).

Long-term affiliation was calculated from data collected as part of routine observations from 2010–2011 (5220 min, described in *de Waal, 1989*), covering approximately one year prior to the start of the experiment. Since the group was highly stable at that time (having been together for more than 30 years), it is unlikely affiliative relationships changed over the course of the experiment. Every 10 minutes a scan sample of affiliative behavior was collected including: grooming, sitting in close proximity, sitting within arm's reach, and play. These data were used to form a sociometric matrix from which adjusted residuals were calculated, a measure comparing observed and expected values (*Everitt, 1977*, Fig. 2). These adjusted residuals were used as a quantitative measure of long-term affiliation in the current study. In addition to routine observations, pant-grunts, a unidirectional submissive vocalization, were recorded *ad libitum* from 2010–2011 and used to determine

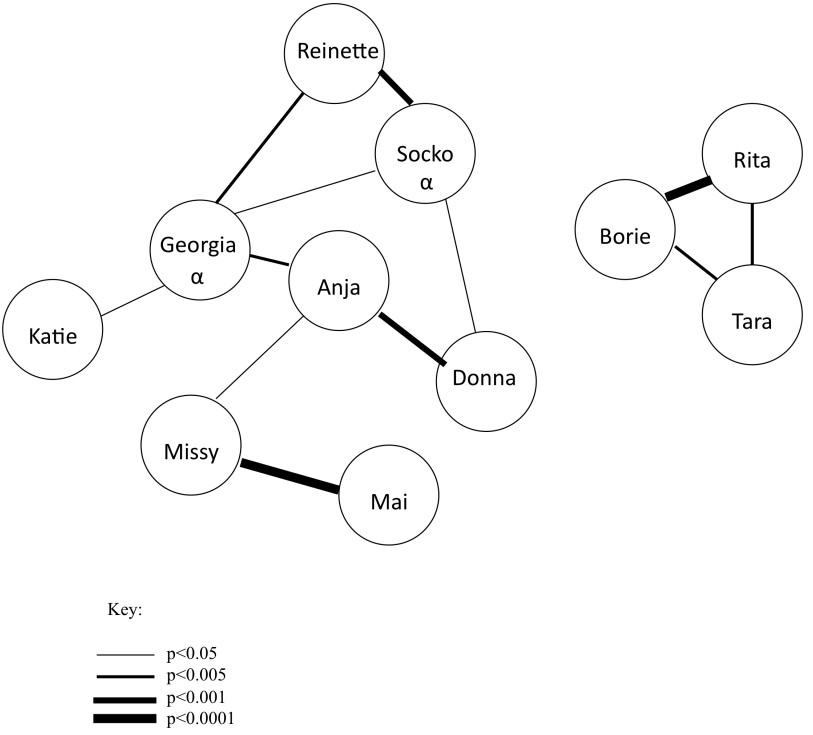

**Figure 2 Affiliation sociogram of the chimpanzee group.** Linkages between individuals indicate a significant, positive association. The thicker the line, the stronger the association between individuals. The alpha male (Socko) and alpha female (Georgia) are denoted with an $\alpha$.

the dominance hierarchy. Although female–female pant grunts were rare, enough were observed during that time period to draw up a linear hierarchy.

## Analyses

### *Understanding of the task*

To test whether the chimpanzees learned about the need for and role of the partner, we compared behavior during the acquisition phases to the proficiency phase. Within-subjects mixed measures ANOVAs were run to compare differences between phases and partner conditions (dyadic and triadic) for latency, efficiency, and pulling. Latency and efficiency (number of pulls to success) were both measured from the time the succeeding pair or triad arrived at the apparatus until the time of success, when they obtained the food. We compared the pulling rate (pulls per minute) of each individual when the correct number of partners was at the bar "ready" to pull versus when there were not enough partners present. This is a very strict criterion as chimpanzees who momentarily stepped away from the apparatus or were approaching but not yet within reach of the bar were not considered to be "present". In the dyadic condition we compared the pulling rate when a partner was present versus absent and in the triadic condition we compared the pulling rate when all three partners were in place to when only two or one partners were in place. We compared these rates between the acquisition and proficiency phases to check for

developing understanding of the need for a partner over time. All pulls, even those that occurred in the absence of success were included in this analysis. All acquisition analyses were run using SPSS Statistics 20.0 (IBM, inc.).

### *Partner choice*

We explored which chimpanzees chose to approach the apparatus when other chimpanzees were already there as potential partners. When a chimpanzee was present at the apparatus, he or she had 10 potential partners that could approach. If another individual approached, they were scored as a 1, whereas individuals who did not approach were scored as 0. This was done by session, so if a chimpanzee was never at the apparatus during a given session, they were excluded from the analysis for that session (since they had to be at the apparatus in order for someone to approach to work with them). All triads were broken down into their corresponding dyads for the purpose of analysis. Partner choice analyses are limited to the proficiency phase only as we could confirm at that point the chimpanzees knew they needed a partner.

To determine which factors influenced partner choice, we ran a generalized linear mixed model (GLMM), with approach as a binomial dependent variable. Kinship, long-term affiliation, rank distance between the two individuals, recent success (total number of successes for that pair during the current phase) and relative past success (the percentage of that individual's success from the previous phases with that partner) were included in various combinations as fixed terms. The advantage to using two different measures of past success is that if past success is the key to partner choice, we can determine whether the chimpanzees gauge success based on cumulative rewards obtained (which could result in partner choice by reinforcement) or if they judge partners based on their relative effectiveness at the task (which would reflect a more sophisticated evaluation of available partners). Models examined each fixed effect independently as well as interactions between the effects. None of the fixed effects were correlated with each other. We also ran a full model which contained all of the fixed effects and a null model that contained only the random effects for all phases. Dyadic and triadic sessions were analyzed separately and in total we examined 14 models for each. Identity of the chimpanzee already present at the apparatus, identity of the chimpanzee that approached, and session were included as random effect to control for repeated sampling, frequency of presence at the apparatus, frequency of approach, and interdependence between dyads. We used an ANOVA to determine which model had the most explanatory power by comparing the Akaike's information criterion (AIC) for all of the possible models. Once the best model was identified, we used a Markov chain Monte Carlo simulation of 10,000 interactions to obtain significance values. All partner choice analyses were run using R statistical software (2012), with the lme4 package.

## RESULTS

### Understanding of the task

For both dyadic and triadic tests, at least four different chimpanzees spontaneously solved the task within the first 2 h of exposure without any training. Overall, 10 of the

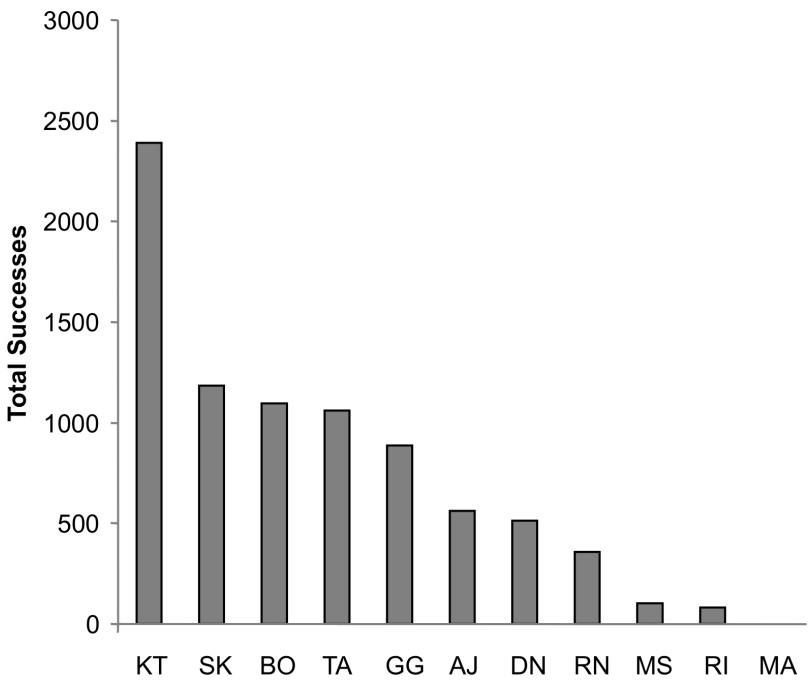

**Figure 3 Success by individual chimpanzee.** The total number of successes for each chimpanzee across all of the test sessions. All individuals except for MA achieved at least 80 successes, with nine chimpanzees achieving over 100.

11 chimpanzees solved the task at least once during both the dyadic and triadic tests for a total of 2,462 dyadic successes and 1,103 triadic successes. By the end of the acquisition phase, the average chimpanzee had succeeded 139 times at the dyadic task and 99 times at the triadic task. This increased to an average success of 447 times at the dyadic task and 301 times at the triadic task by the end of the proficiency phase. The total number of successes by each individual across the entire study is displayed in Fig. 3. One female, Mai, failed to solve the task in either dyadic or triadic tests and in fact ceased making pulling attempts before the proficiency phase. Since her overall pulling rates were more than two standard deviations below the group mean, she was eliminated from the analysis.

Latency to success was significantly lower in the proficiency phase than in the acquisition phase (Mixed Measures ANOVA: $F_{1,7} = 21.29, p = 0.002$). Similarly, extraneous pulling, i.e., pulls that did not lead to success, dropped significantly from the acquisition to the proficiency phase. The chimpanzees succeeded with significantly fewer pulls per success (e.g., higher efficiency) during the proficiency phase (Mixed Measures ANOVA: $F_{1,7} = 16.83, p = 0.005$, Fig. 4). For both of these measures there was no significant difference between dyadic and triadic tests (latency: $F_{1,7} = 0.11, p = 0.75$; extra pulling: $F_{1,7} = 0.18, p = 0.68$), demonstrating increased effectiveness of cooperation regardless of how many chimpanzees were needed for the task. There were, however, significant interactions (latency: $F_{1,7} = 13.95, p = 0.007$; extra pulling: $F_{1,7} = 10.37, p = 0.015$), demonstrating a larger change from dyadic acquisition to proficiency than from triadic acquisition to proficiency. It is important to note, however, that these phases for dyads were

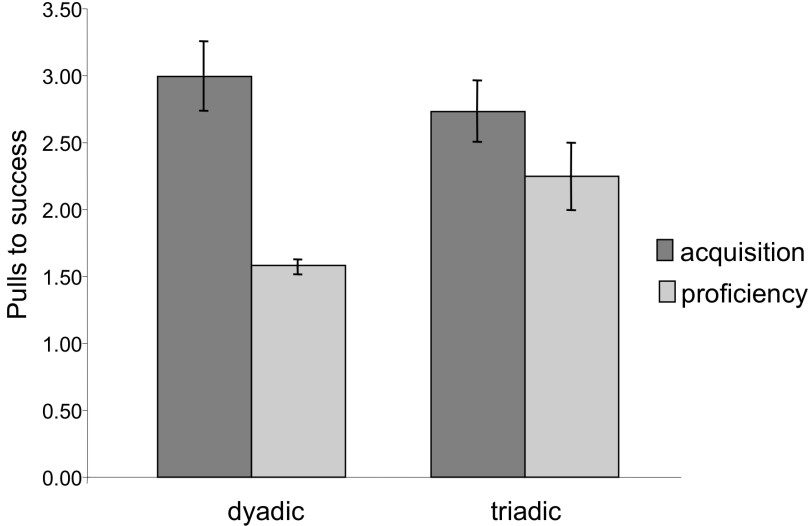

**Figure 4 Number of pulls to success during the acquisition and proficiency phases.** Extraneous pulling that did not lead to success decreased between the two phases for both dyadic and triadic partners. There was no significant difference between the dyadic and triadic conditions.

separated in time by the triadic acquisition phase, whereas the proficiency phase for triads immediately followed the triadic acquisition phase.

In order to assess whether the chimpanzees developed an understanding of the need for a partner, we compared pulling rates when the correct number of individuals were present and sitting ready at the bars to pull to when an insufficient number was present. In the dyadic test sessions there was a significant effect of partner presence; chimpanzees pulled more when a partner was at the other bar then when no partner was present (Mixed Measures ANOVA: $F_{1,9} = 39.53, p < 0.001$; Fig. 5). There was also a significant phase by partner presence interaction: the ratio of pulls when a partner was present as compared to pulls when a partner was absent was greater in the proficiency phase than in the acquisition phase ($F_{1,9} = 14.11, p = 0.005$). Finally, there was an overall effect of phase, such that individuals had higher overall pulling rates in the proficiency phase than in the acquisition phase, however this increase was primarily observed when a partner was present ($F_{1,9} = 9.76, p = 0.01$).

In triadic tests, the trends were similar but less pronounced. There was a main effect of partner presence: when two other partners were present the chimpanzees pulled more than when there was only one or zero partners present ($F_{1.12,10.99} = 11.62, p = 0.006$, Greenhouse–Geisser corrected due to lack of sphericity). Unlike dyadic tests, however, the phase by number of partners interaction was no longer significant ($F_{1.07,9.59} = 0.962$, $p = 0.36$, Greenhouse–Geisser corrected due to lack of sphericity) and there was no longer a main effect of phase ($F_{1,9} = 2.49, p = 0.15$).

Most of the chimpanzees spontaneously developed a bias for a particular position at the apparatus. In the dyadic task, three chimpanzees had significantly more success (as determined by a binomial test $p < 0.05$) at the barrier than the tray position, two chimpanzees had no preference and five chimpanzees had significantly more success at

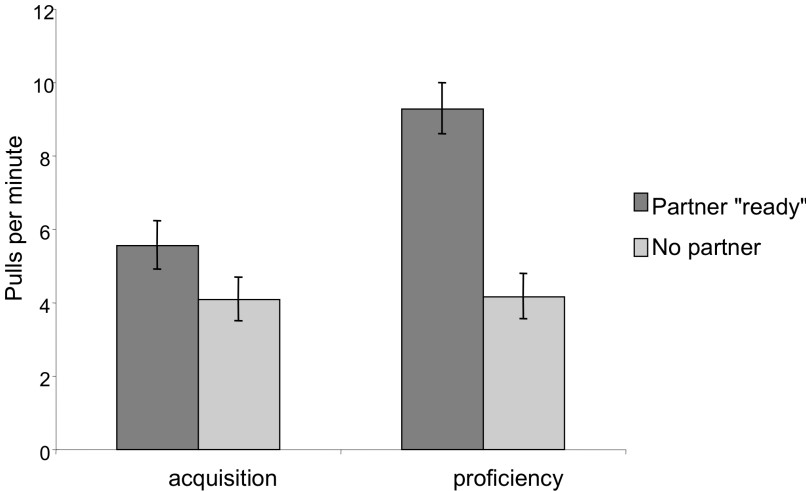

**Figure 5 Comparison of pulling rates per minute during the acquisition and proficiency phases for dyadic cooperation.** Partner "ready" indicates that a partner was both present at the apparatus in a position to be able to pull on the tray. The pattern was similar for triadic pulling rates.

the tray than the barrier position. In the triadic condition, there was more skew towards barrier preference, likely due to the fact that there were two barrier positions available and only one tray position.

## Partner choice

There were a total of 45 possible dyadic combinations and 120 possible triadic combinations among all of the chimpanzees who solved the task ($N = 10$). In total, 29 unique dyads and 32 unique triads manifested actual cooperation in the experiments.

Applying GLMM, we examined factors that might influence whether or not one individual approaches another already present at the apparatus. Across both dyadic and triadic tests, there was a significant influence of the tested random effects (intercept; dyadic: $Z = -4.30, p < 0.001$; triadic: $Z = -5.11, p < 0.001$; random effects included identity of the approacher and individual already there and test session number). Given the high variability of individual participation in the task, it is not surprising that most of the variance of the random effects comes from individual identities rather than the session number (Table 2).

Once individual identity was controlled for, the same model had the best fit for both dyadic and triadic sessions. It included kinship, affiliation as well as the interaction of these factors with rank distance (dyadic: AIC $= 601.44$, $\chi^2 = 9.68$, $df = 0$, $p < 0.001$; triadic: AIC $= 1199.22$, $\chi^2 = 4.12$, $df = 0$, $p < 0.001$; Table 2). The interaction between kinship and rank distance was significant (dyadic sessions: $Z = 3.80, p < 0.001$; triadic sessions: $Z = 2.67, p = 0.007$), reflecting the reluctance of chimpanzees to approach individuals much higher ranking than themselves, unless these individuals were relatives. Additionally, for the dyadic sessions, there were main effects of kinship ($Z = -2.07, p = 0.04$) and rank distance ($Z = -3.86, p < 0.001$) but these effects were not found in triadic sessions (kinship: $Z = -1.05, p = 0.30$; rank distance: $Z = -.58, p = 0.56$). Affiliation and the

**Table 2  Results of the best fit GLMM during the proficiency phase.** Fixed effects in bold had a significant influence on whether or not an individual approached. In both dyadic and triadic sessions, individuals were more likely to approach others close in rank to themselves, unless the potential partner was kin.

| Variable | $\beta$ | SE | Z | p |
|---|---|---|---|---|
| *Dyadic proficiency* | | | | |
| Fixed effects | | | | |
| Intercept | −1.98 | 0.46 | −4.30 | <0.001 |
| **Kin** | **−1.38** | **0.67** | **−2.07** | **0.04** |
| Affiliation | 0.12 | 0.10 | 1.22 | 0.22 |
| **Rank distance** | **−0.32** | **0.08** | **−3.86** | **<0.001** |
| **Kin ∗ rank distance** | **0.52** | **0.13** | **3.80** | **<0.001** |
| Affiliation ∗ rank distance | −0.03 | 0.03 | −1.12 | 0.26 |
| Random effects | | | | |
| Individual present | Variance | 0.26 | | |
| Individual approaching | Variance | 0.23 | | |
| Session | Variance | 0.00 | | |
| *Triadic proficiency* | | | | |
| Fixed Effects | | | | |
| **Intercept** | **−1.79** | **0.35** | **−5.11** | **<0.001** |
| Kin | −0.47 | 0.45 | −1.05 | 0.30 |
| Affiliation | −0.02 | 0.05 | −.26 | 0.80 |
| Rank distance | −0.02 | 0.04 | −0.58 | 0.56 |
| **Kin ∗ rank distance** | **0.22** | **0.08** | **2.67** | **0.007** |
| Affiliation ∗ rank distance | 0.02 | 0.01 | 1.26 | 0.21 |
| Random effects | | | | |
| Individual present | Variance | 0.21 | | |
| Individual approaching | Variance | 0.19 | | |
| Session | Variance | 0.08 | | |

interaction between affiliation and rank distance were not significant in either the dyadic or triadic sessions. Since the best fit model did not include any measure of past success, we examined a full model to see if past success was playing any role in approach. There was no significant effect of recent success or all past success in either the dyadic (recent success: $Z = 0.73, p = 0.46$; all past success: $Z = −1.50, p = 0.13$) or triadic (recent success: $Z = 1.74, p = 0.08$; all past success: $Z = 0.35, p = 0.72$) proficiency phase.

The effect of rank distance indicates that individuals of similar rank were likely to approach each other. There was no overall effect of rank on task performance, i.e., individuals of high rank did not have more successes than individuals of lower rank (Spearmans' rank correlation between individual rank and number of successes; $r_s = 0.07$, $N = 10$ individuals, $p = 0.44$).

Finally, approaches tended to be reciprocal—that is, the more frequently individual A approached B at the apparatus, the more frequently B approached A (dyadic: $r_s = 0.42$, $N = 90, p < 0.001$; triadic: $r_s = 0.56, N = 90, p < 0.001$). Note that the $p$-values reported

here are exact two-tailed *p*-values obtained from 10,000 random permutations so as to address interdependence between dyads.

## DISCUSSION

Without any specific training, the chimpanzees in this study spontaneously solved the cooperation task and were extremely successful under both dyadic and triadic conditions. The high success rate, with a total of 3,565 completed cooperative acts (an average of 38 per one hour test session), confirms observations of cooperation in nature: chimpanzees are capable of cooperating in more complex open environments than typically tested. The chimpanzees were clearly highly motivated to participate in the task. Since the number of successes during the proficiency phase was higher in both the dyadic and triadic conditions, it is unlikely the chimpanzees habituated to the task. Furthermore, the task was run over 10-month period with an average of 2–3 sessions per week, to prevent both habituation and a loss of motivation from too frequent testing.

The current study contrasts with previous work in a number of ways. First, in many studies the chimpanzees required extensive training (*Crawford, 1937*), or had been individually familiarized with the apparatus before any cooperative testing (*Melis, Hare & Tomasello, 2006a*; *Melis, Hare & Tomasello, 2006b*). In the only previous study without pre-training, 5 out of the 6 chimpanzees showed no understanding of the task and were just as likely to pull when a partner was present versus absent (*Chalmeau, 1994*). In the current study, the chimpanzees had no experience with a pulling apparatus of any kind prior to the dyadic acquisition phase. However, it might be argued that the dyadic acquisition phase served as pre-training for the triadic phase. If so, we would expect to see high rates of pulling when only one other partner was present in the triadic phase of testing. This was not the case, however: the chimpanzees pulled the most when both partners were present, less when one partner was present and the least was when no other partner was present. The low frequency of pulling when an insufficient number of partners were present demonstrates an understanding of the triadic nature of the task, which manifested itself right at the beginning of the triadic acquisition phase.

Despite the chimpanzees' demonstrable sensitivity to partner presence in both the dyadic and triadic phase, pulling in the absence of a needed partner never fully disappeared. Incomplete extinction of such pulling was probably due to continuing conditioning effects as well as the conservative measure of partner presence employed: the partner had to be at the bar, ready to pull. Therefore, any pulls made as a partner was approaching or nearby were counted as pulling when a partner was "not ready". It is possible that the chimpanzees viewed a partner approach as a signal to start the task. Moreover, pulling was an extremely low cost behavior. The energy expended on pulling might simply not have been great enough to deter extraneous pulling.

One of the surprises of this study was the high level of success without any pre-training. Previous work has shown that more intuitive tasks, where individuals are pulling food towards them (e.g., *Mendres & de Waal, 2000*), are learned faster and showed greater understanding than non-intuitive tasks where pulling is not mechanically connected to food

delivery. Thus primates participating in weighted tray or string-pulling tasks (*Hirata & Fuwa, 2007*; *Melis, Hare & Tomasello, 2006a*; *Melis, Hare & Tomasello, 2006b*; *Mendres & de Waal, 2000*) have had more success than those participating in lever-pressing (*Chalmeau, Visalberghi & Gallo, 1997*; *Visalberghi, Quarantotti & Tranchida, 2000*). One exception to this is *Crawford*'s (*1937*) original weighted tray task, which required extensive training. However, Crawford's chimpanzees were juveniles and in later experiments (including the current study) the participants were all adults. Additionally, the chimpanzees in Crawford's study did show an understanding of the partner's role: one chimpanzee would recruit the other to help him. Since the chimpanzees in the current study could clearly see the mechanical results of their actions and how their actions resulted in food delivery it is not surprising they developed an understanding of the task.

Another unexpected finding was how highly successful the apes were despite the group setting in which they were operating. The potential for competition and free-loading did not seem to deter them. Previous studies have demonstrated obstacles to cooperation under free choice conditions due to a lack of inter-individual tolerance (*Burton, 1977*; *Chalmeau, 1994*; *Chalmeau & Gallo, 1996*; *Chalmeau, Visalberghi & Gallo, 1997*; *Fady, 1972*; *Petit, Desportes & Thierry, 1992*). However, in all of these studies the reward for cooperation was a single monopolizable food reward. For most participants in the task, there was no net gain. In the current study, in contrast, each individual received their own reward, resulting in a net gain for all participants. Although this design did not allow us to ask how reward division might influence future partner choice, ensuring a net gain for all participants is the essence of mutualism, which allowed us to examine the details of partner choice.

Kinship and rank similarity were the best predictors of partner choice: the chimpanzees tended to approach individuals of similar rank to themselves unless the individual at the apparatus was their kin. Interestingly, this did not only apply to low ranking individuals approaching other low ranking individuals; high-ranking individuals also preferred approaching high-ranking individuals. Closeness in rank and kinship probably foster partnerships in which competition is mitigated (*de Waal, 1986*; *de Waal & Luttrell, 1986*; *Silk, 1982*). These partnerships are characterized by higher social tolerance than ones with large discrepancies in rank, which often results in the higher-ranking individual forcefully claiming food. Our results are consistent with previous studies that reported higher levels of cooperation between tolerant individuals (*de Waal & Davis, 2003*; *Melis, Hare & Tomasello, 2006a*; *Petit, Desportes & Thierry, 1992*). In one previous study of partner choice, the alpha male monopolized the apparatus and rewards, resulting in a lack of interest of the group to approach while he was there (*Chalmeau, 1994*). In the current study, there was a high level of participation by 10 out of the 11 individuals present, acting in a wide variety of partnerships. The alpha male participated, but without excluding others. His most frequent partners were middle- to high-ranking females, i.e., females fairly close to his own rank.

Although we only had one male in the group, which limited opportunities to examine male–male cooperation, the number of females that participated in the task allowed

us to examine the dynamics of female–female cooperation, which have gone largely understudied. Although wild chimpanzee females rarely cooperate due to ecological constraints, in our study females spontaneously cooperated in both female–female and mixed sex dyads and triads. The females in our study demonstrated the high potential for female–female cooperation and allowed us to investigate the dynamics of female partner choice.

Further evidence of the high level of social tolerance between partners is demonstrated by the low rate of agonism observed throughout the study. Agonism was extremely rare, occurring in only about 1% of all trials. Escalated agonism (e.g., slapping, biting, or grabbing) was rarer still, occurring in only 0.1% of trials.

The emphasis on tolerant partnerships means that the chimpanzees were not choosing the most successful partners available. These results, combined with previous work (*Burton, 1977*; *Chalmeau, 1994*; *Chalmeau & Gallo, 1996*; *Chalmeau, Visalberghi & Gallo, 1997*; *Fady, 1972*; *Petit, Desportes & Thierry, 1992*) seem to suggest that while theoretically the chimpanzees should choose the most successful individuals to maximize their own gain, there may be social constraints on their ability to display this tendency. Indeed, when social constraints are taken away by limiting partner choice to only two individuals who were socially tolerant, chimpanzees did choose the most successful partners (*Melis, Hare & Tomasello, 2006b*). From an evolutionary standpoint, social relationships are long-term investments that encompass a variety of interactions (including grooming, agonistic support, sex, play, and food sharing). Cooperation is only one of many different currencies being exchanged in a marketplace. Rather than being "irrational", choosing a tolerant partner may reflect the most economical choice: a safe investment that is likely to lead to equal outcomes for all participants, in the present and in future interactions.

## ACKNOWLEDGEMENTS

We would like to thank Victoria Horner, Darby Proctor, Zanna Clay, Harold Gouzoules, Sarah Brosnan, Monica Capra, and Philippe Rochat for helpful discussions; Julia Watzek for statistical help; the Veterinary and Animal Care staff at the Yerkes National Primate Research Center for maintaining the health of our research subjects. The YNPRC is fully accredited by the American Association for Accreditation for Laboratory Animal Care (AAALAC).

### Funding

The study was supported by the Living Links Center, Emory's PRISM Program (NSF GK12 #DGE0536941), Emory's Dean's Teaching Fellowship program, Emory's FIRST program (NIH/NIGMS(USA) IRACDA grant #K12GM00680 to MWC), the Expanding the Science and Practice of Gratitude Project of the Greater Good Science Center at the University of California-Berkeley, and the base grant of the National Institutes of Health to the YNPRC from the National Center for Research Resources PR51RR165 (currently supported by the Office of Research Infrastructure Programs/ODP51OD11132). The funders had no role

in study design, data collection and analysis, decision to publish, or preparation of the manuscript.

## Grant Disclosures

The following grant information was disclosed by the authors:

Emory's PRISM Program (NSF GK12): #DGE0536941.

Emory's FIRST program (NIH/NIGMS(USA) IRACDA grant): #K12GM00680.

Greater Good Science Center, University of California-Berkeley.

National Center for Research Resources: PR51RR165.

Office of Research Infrastructure Programs: ODP51OD11132.

## Competing Interests

The authors declare there are no competing interests.

## Author Contributions

- Malini Suchak conceived and designed the experiments, performed the experiments, analyzed the data, wrote the paper, prepared figures and/or tables, reviewed drafts of the paper.
- Timothy M. Eppley and Matthew W. Campbell conceived and designed the experiments, performed the experiments, reviewed drafts of the paper.
- Frans B.M. de Waal conceived and designed the experiments, analyzed the data, wrote the paper, prepared figures and/or tables, reviewed drafts of the paper.

## Animal Ethics

The following information was supplied relating to ethical approvals (i.e., approving body and any reference numbers):

Emory University Institutional Animal Care and Use Committee (IACUC), protocol #YER-2000180-53114GA.

## Supplemental Information

Supplemental information for this article can be found online at http://dx.doi.org/10.7717/peerj.417.

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
