# Peer review of "Ape duos and trios: spontaneous cooperation with free partner choice in chimpanzees"

_PeerJ, doi:10.7717/peerj.417_

## Round 0.1 · original submission · Major Revisions

Please consider all of the comments made by the reviewers and make revisions to your manuscript accordingly.

·

Basic reporting

No Comments

Experimental design

I have a few questions regarding the fact that the data on long-term affiliation was collected in 2010-2011, whether it is valid.
When did you run the cooperation experiment? When was the group of the subject chimpanzees formed? Is the group stable?
If the group was formed recently, the group was not really stable, and the experiment was carried out sometime after 2011, then data from 2010-2011 on the chimpanzee relationships may not be valid.
The group stability is a kind of subjective judgment, but at least I think the authors should show objective data regarding when the experiment was done, when the group was formed, and whether there was no change in dominance rank.
Regarding the dominance rank, the authors stated that pant-grunts were used to determine the rank. However, to my knowledge, females generally do not emit pant-grants to another female. I guess the authors used other behavioral criteria to determine the rank among females.

Validity of the findings

The authors explained "there was no significant difference between dyadic and triadic tests" concerning the latency to success and extra pulling.
I wonder if this is really true.
Judging from the bars and error bars of Figure 3, it seems that there was an interaction between phase (acquisition vs. proficiency) and condition (dyadic vs. triadic) and that there is a significant difference between dyadic and triadic test conditions in proficiency phase.
Assessing just a simple main effect of phase may not be appropriate.
I would like to see more detailed statistical results.

The authors state in the Discussion that "the chimpanzees were not choosing the most successful partners available".
However, the results supporting this notion are not described in the Results section.
I would like to see the results concerning the effects of "recent success" and "past success" in the Results section.

Additional comments

Line 303, "Table 1". This should be Table 2.

I think my questions and comments are rather minor ones, and that the manuscript is well written in general.

Reviewer 2 ·

Basic reporting

This paper is a delight. It tests cooperation in chimpanzees and does so in an astute and considered manner. For decades we have endured and lived within the framework that all life had to be selfish and competitive and this paper convincingly demonstrates that some cooperative behaviour does not require training and is already part of the behavioral repertoire of the species. For the thought and outcome alone this paper is worth publishing.

Experimental design

In general, the design seems flawless, clever and simple. There are a few questions, however, which need to be answered by adding explanations in the method section.
Q1 re Number of Sessions: some clarification is needed for the statement in lines 161-165: if a session was 1 hour per day, the sentence of lines 163-4 is ambiguous—if dominant individuals hawked the apparatus early, then the sessions were extended? How can one understand the oblique ‘longer session’? does it mean that the time one individual had monopolized the apparatus is substracted from the total? –It is implied in this sentence that sessions actually exceeded 1 hour and if so, how often did this happen? And the sessions would become a little arbitrary, right? That is, the protocol was broken. If this reader misunderstood this, please consider rewriting this sentence because other readers could also be confused by this sentence.
Q2 re sex of individuals used: there is a sentence in Line 391 that expresses regret that only one male could be used. I would not stress a ‘lack’ in the paper but rather think that the researchers should make the selection of females clear as a strength in the beginning of the method section and give some good reasons why testing females may be very important. It certainly is important and can make a contribution to our understanding of how group dynamics work in chimpanzee groups. For instance, if memory serves me right, alpha males tend to choose low ranking individuals to share spoils with him in order to increase allegiance and his power base but specifically exclude those close in rank or kin. Hence, the new findings of this paper partly rest on sex differences. It seems particularly important to emphasise that female cooperation exists but largely at same status levels and argue the point right at the beginning and throughout the paper rather than raise the topic as an apology at the end (Line 391). I do not understand why the use of females and the results are, in that sense, so underplayed in the paper.

Validity of the findings

The findings have been subjected to very detailed and sophisticated statistical procedures and seem solid and trustworthy.

Additional comments

The paper is basically excellent but there are weaknesses by omission. I believe that the points may be easily fixed but need addressing.
Point 1-- has been raised under methods—the sex differences are important but as the paper stands it almost tries to blur that dimension of the paper and yet this may well be one of its strengths and should be turned into a strength.
Point 2-- most definitely required is an explanation as to why the chimpanzees showed such an exceptionally high level of cooperation with the researchers. The paper states that 3,565 acts of cooperation were documented during the experiments. This, as they state, amounts to 38 cooperative acts per hour— meaning roughly one per every 1 min and 50secs.
The question that arises and needs addressing is about motivation. Why were the females so keen to extract the small food items at all and, more importantly, at such a high level of retrieval. What was it that might explain the extraordinarily high level of motivation? The subjects, as I understood, were well fed and fed at predictable and regular intervals. The treats seemed rather average and ordinary to me even if exchanged (meaning they comprised food items generally part of a normal diet).
I am simply not convinced that these tit bits by themselves can explain the high level of voluntarism because the paper fails to explain the chimpanzees’ singular focus on the task. This point might make readers rather skeptical and further explanation could remove any reservations.

Point 3--It would also help to provide a figure of an ethogram of the females, showing the highest ranking female and her connections and most frequent interactions with others, also indicating rank and kin relation. There could be an effect of status on actions and that might partly explain the high incidents of task performance of the chimps. The researchers obviously have all the data because they carefully noted several categories of behavior including those that were affiliative.

Point 4—If there were 3,565 acts of cooperation recorded and a day’s session was just one hour (with 38 cooperative acts) -then the experiments were run over roughly 100 days –was that every day? If so, what is possibility of habituation –again this also comes back to motivation and a sentence in the discussion could clarify those points and it would relieve all the angst if the experiments were run over a longer time period. If this is stated somewhere in the paper and this reader has overlooked this-sincere apologies but it could also mean that others would respond in the same way and that is worth avoiding.

Point 5-- As asked before--were all sessions 1 hour long or were the sessions extended when one chimpanzee took hold of the apparatus

Point 6—a minor point -- Line 377 speaks of ‘rank distance’ when it really refers to rank similarity—using the latter would be less confusing although it is understood that ‘rank distance’ was used as a unit of measurement-in this case it does not sit well in this sentence. .

---

## Round 0.2 · accepted · Accept

Thank you for revising your manuscript in accordance with the reviewers' suggestions.

Reviewer 2 ·

Basic reporting

This paper has been read and reviewed by me before. I have now read the resubmission, the rebuttal and have noted the changes the authors have made. The revised copy was a pleasure to read. The authors have taken great care in clarifying the points identified previously and have removed any concerns raised. It is now a very convincing and important paper and I would verh much support its publication. Congratulations to the authors for this fine paper and their original contribution to our understanding of the nature and possibilities cooperation in animals.

Experimental design

excellent- no further comment (see above)

Validity of the findings

very convincing

Additional comments

as per above